# DEEP LEARNING WITH DATA PRIVACY VIA RESIDUAL PERTURBATION

## ABSTRACT

Protecting data privacy in deep learning (DL) is at its urgency. Several celebrated privacy notions have been established and used for privacy-preserving DL. However, many of the existing mechanisms achieve data privacy at the cost of significant utility degradation. In this paper, we propose a stochastic differential equation principled *residual perturbation* for privacy-preserving DL, which injects Gaussian noise into each residual mapping of ResNets. Theoretically, we prove that residual perturbation guarantees differential privacy (DP) and reduces the generalization gap for DL. Empirically, we show that residual perturbation outperforms the state-of-the-art DP stochastic gradient descent (DPSGD) in both membership privacy protection and maintaining the DL models' utility. For instance, in the process of training ResNet8 for the IDC dataset classification, residual perturbation obtains an accuracy of 85.7% and protects the perfect membership privacy; in contrast, DPSGD achieves an accuracy of 82.8% and protects worse membership privacy.

## 1 INTRODUCTION

Many high-capacity deep nets (DNs) are trained with private data, including medical images and financial transaction data (Yuen et al., 2011; Feng et al., 2017; Liu et al., 2017). DNs usually overfit and can memorize the private training data, which makes training DNs exposed to data privacy leakage (Fredrikson et al., 2015a; Shokri et al., 2017; Salem et al., 2018; Yeom et al., 2018; Sablayrolles et al., 2018). Given a pre-trained DN, the membership inference attack can determine if an instance is in the training set based on DN's response (Fredrikson et al., 2014; Shokri et al., 2017; Salem et al., 2018); the model extraction attack can learn a surrogate model that matches the target model, given the adversary only black-box access to the target model (Tramèr et al., 2016; Gong & Liu, 2018); the model inversion attack can infer certain features of a given input from the output of a target model (Fredrikson et al., 2015b; Al-Rubaie & Chang, 2016); the attribute inference attack can deanonymize the anonymized training data (Gong & Liu, 2016; Zheng et al., 2018).

Machine learning (ML) with data privacy is crucial in many applications (Lindell & Pinkas, 2000; Barreno et al., 2006; Hesamifard et al., 2018; Bae et al., 2019). Several algorithms have been developed to reduce privacy leakage include differential privacy (DP) (Dwork et al., 2006), federated learning (FL) (McMahan et al., 2016; Konečný et al., 2016), and $k$-anonymity (Sweeney, 2002; El Emam & Dankar, 2008). Objective, output, and gradient perturbations are among the most used approaches for ML with DP guarantees at the cost of significant utility degradation (Chaudhuri et al., 2011; Bassily et al., 2014; Shokri & Shmatikov, 2015; Abadi et al., 2016b; Bagdasaryan et al., 2019). FL trains centralized ML models, through gradient exchange, with the training data being distributed at the edge devices. However, the gradient exchange can still leak the privacy (Zhu et al., 2019; Wang et al., 2019c). Most of the existing privacy is achieved at a tremendous sacrifice of utility. Moreover, training ML models using the state-of-the-art DP stochastic gradient descent (DPSGD) leads to tremendous computational cost due to the requirement of computing and clipping the per-sample gradient (Abadi et al., 2016a). It remains a great interest to develop new privacy-preserving ML algorithms without excessive computational overhead or degrading the utility of the ML models.

### 1.1 OUR CONTRIBUTION

In this paper, we propose *residual perturbation* for privacy-preserving deep learning (DL) with DP guarantees. At the core of residual perturbation is injecting Gaussian noise to each residual mapping

of ResNet (He et al., 2016), and the residual perturbation is theoretically principled by the stochastic differential equation (SDE) theory. The major advantages of residual perturbation are threefold:

- It can protect the membership privacy of the training data almost perfectly and often without sacrificing ResNets' utility. Furthermore, it can even improve ResNets' classification accuracy.
- It has fewer hyperparameters to tune than the benchmark DPSGD. Also, it is more computationally efficient than DPSGD, which requires to compute the per-sample gradient.
- It can be easily implemented by a few lines of code in modern DL libraries.

## 1.2 RELATED WORK

Improving the utility of ML models with DP guarantees is an important task. PATE (Papernot et al., 2017; 2018) uses semi-supervised learning together with model transfer between the "student" and "teacher" models to enhance utility. Several variants of the DP notions have also been proposed to improve the privacy budget and some times can also improve the resulting model's utility at a given DP budget (Abadi et al., 2016b; Mironov, 2017; Wang et al., 2018; Dong et al., 2019). Some post-processing techniques have also been developed to improve the utility of ML models with negligible computational overhead (Wang et al., 2019a; Liang et al., 2020). From the SDE viewpoint, (Li et al., 2019; Wang et al., 2015) showed that several stochastic gradient Monte Carlo samplers could reach state-of-the-art performance in terms of both privacy and utility in Bayesian learning.

Gaussian noise injection in residual learning has been used to improve the robustness of ResNets (Rakin et al., 2018; Wang et al., 2019b; Liu et al., 2019). In this paper, we inject Gaussian noise to each residual mapping to achieve data privacy instead of adversarial robustness.

## 1.3 ORGANIZATION

We organize this paper as follows: In Section 2, we introduce the residual perturbation for privacy-preserving DL. In Section 3, we present the generalization and DP guarantees for residual perturbation. In Section 4, we numerically verify the efficiency of the residual perturbation in protecting data privacy without degrading the underlying models' utility. We end with some concluding remarks. Technical proofs and some more experimental details and results are provided in the appendix.

## 1.4 NOTATIONS

We denote scalars by lower or upper case letters; vectors/ matrices by lower/upper case bold face letters. For a vector $\boldsymbol{x} = (x_1, \cdots, x_d) \in \mathbb{R}^d$, we use $\|\boldsymbol{x}\|_2 = (\sum_{i=1}^d |x_i|^2)^{1/2}$ to denote its $\ell_2$ norm. For a matrix $\mathbf{A}$, we use $\|\mathbf{A}\|_2$ to denote its induced norm by the vector $\ell_2$ norm. We denote the standard Gaussian in $\mathbb{R}^d$ as $\mathcal{N}(\mathbf{0}, \mathbf{I})$ with $\mathbf{I} \in \mathbb{R}^{d \times d}$ being the identity matrix. The set of (positive) real numbers is denoted as $(\mathbb{R}^+) \mathbb{R}$. We use $B(\mathbf{0}, R)$ to denote the ball centered at $\mathbf{0}$ with radius $R$.

## 2 ALGORITHMS

### 2.1 DEEP RESIDUAL LEARNING AND ITS CONTINUOUS ANALOGUE

Given the training set $S_N := \{\boldsymbol{x}_i, y_i\}_{i=1}^N$, with $\{\boldsymbol{x}_i, y_i\} \subset \mathbb{R}^d \times \mathbb{R}$ being a data-label pair. For a given $\boldsymbol{x}_i$ the forward propagation of a ResNet with $M$ residual mappings can be written as

$$\boldsymbol{x}^{l+1} = \boldsymbol{x}^l + \hat{F}(\boldsymbol{x}^l, \mathbf{W}^l), \text{ for } l = 0, 1, \cdots, M-1, \text{ with } \boldsymbol{x}^0 = \boldsymbol{x}_i; \ \hat{y}_i = f(\boldsymbol{x}^M), \quad (1)$$

where $\hat{F}(\cdot, \mathbf{W}^l)$ is the nonlinear mapping of the $l$th residual mapping parameterized by $\mathbf{W}^l$; $f$ is the output activation function, and $\hat{y}_i$ is the predicted label for $\boldsymbol{x}_i$. The heuristic continuum limit of (1) is

$$d\boldsymbol{x}(t) = F(\boldsymbol{x}(t), \mathbf{W}(t))dt, \ \boldsymbol{x}(0) = \hat{\boldsymbol{x}}, \text{ where } t \text{ is the time variable.} \quad (2)$$

The ordinary differential equation (ODE) (2) can be revertible, and thus the ResNet counterpart might be exposed to data privacy leakage. For instance, we use the ICLR logo (Fig. 1 (a)) as the initial data $\hat{\boldsymbol{x}}$ in (2). Then we simulate the forward propagation of ResNet by solving (2) from $t = 0$ to $t = 1$ using the forward Euler solver with a time step size $\Delta t = 0.01$ and a given velocity field $F(\boldsymbol{x}(t), \mathbf{W}(t))$ (see Appendix E for the details of $F(\boldsymbol{x}(t), \mathbf{W}(t))$), which maps the original image to its features (Fig. 1 (b)). To recover the original image, we start from the feature and use the backward Euler iteration, i.e., $\tilde{\boldsymbol{x}}(t) = \tilde{\boldsymbol{x}}(t + \Delta t) - \Delta t F(\tilde{\boldsymbol{x}}(t + \Delta t), t + \Delta t)$, to evolve $\tilde{\boldsymbol{x}}(t)$ from $t = 1$ to $t = 0$ with $\tilde{\boldsymbol{x}}(1) = \boldsymbol{x}(1)$ being the features obtained in the forward propagation. We plot the recovered image from features in Fig. 1 (c), and the original image can be almost perfectly recovered.

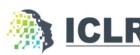 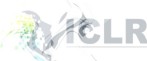 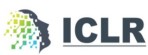 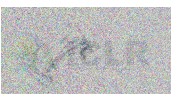 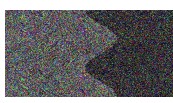

(a) $\boldsymbol{x}(0)$ (ICLR logo)  (b) $\boldsymbol{x}(1)$ (ODE )  (c) $\tilde{\boldsymbol{x}}(0)$ (ODE )  (d) $\boldsymbol{x}(1)$ (SDE )  (e) $\tilde{\boldsymbol{x}}(0)$ (SDE )

Figure 1: Illustrations of the forward and backward propagation of the training data using 2D ODE (2) and SDE (3) models. (a) the original image; (b) & (d) the features of the original image generated by the forward propagation using ODE and SDE, respectively; (c) & (e) the recovered images by reverse-engineering the features shown in (b) & (d), respectively. We see that it is easy to break the privacy of the ODE model, but harder for SDE.

## 2.2 RESIDUAL PERTURBATION AND ITS SDE ANALOGUE

In this part, we propose two SDE models to reduce the reversibility of (2), and the corresponding residual perturbations analogue can protect the data privacy in DL.

**Strategy I.**  For the first strategy, we consider the following SDE model:

$$d\boldsymbol{x}(t) = F(\boldsymbol{x}(t), \mathbf{W}(t))dt + \gamma d\mathbf{B}(t),\ \gamma > 0, \tag{3}$$

where $\mathbf{B}(t)$ is the standard Brownian motion. We simulate the forward propagation and reverse-engineering the input from the output by solving the SDE model (3) with $\gamma = 1$ using the same $F(\boldsymbol{x}(t), \mathbf{W}(t))$ and initial data $\hat{\boldsymbol{x}}$. We use the following forward (4) and backward (5) Euler-Maruyama discretizations (Higham, 2001) of (3),

$$\boldsymbol{x}(t + \Delta t) = \boldsymbol{x}(t) + \Delta t F(\boldsymbol{x}(t), \mathbf{W}(t)) + \gamma \mathcal{N}(\mathbf{0}, \sqrt{\Delta t}\,\mathbf{I}), \tag{4}$$

$$\tilde{\boldsymbol{x}}(t) = \tilde{\boldsymbol{x}}(t + \Delta t) - \Delta t F(\tilde{\boldsymbol{x}}(t + \Delta t), \mathbf{W}(t + \Delta t)) + \gamma \mathcal{N}(\mathbf{0}, \sqrt{\Delta t}\,\mathbf{I}), \tag{5}$$

for the forward and backward propagation, respectively. Figure 1 (d) and (e) show the results of the forward and backward propagation by SDE, respectively, and these results show that it is much harder to reverse the features obtained by SDE evolution. The SDE model informs us to inject Gaussian noise, in both training and test phases, to each residual mapping of ResNet to protect data privacy, which results in

$$\boldsymbol{x}^{i+1} = \boldsymbol{x}^i + \hat{F}(\boldsymbol{x}^i, \mathbf{W}^i) + \gamma \boldsymbol{n}^i,\ \text{where } \boldsymbol{n}^i \sim \mathcal{N}(\mathbf{0}, \mathbf{I})^1. \tag{6}$$

**Strategy II.**  For the second strategy, we consider using the multiplicative noise instead of the additive noise used in (3) [2] and (6), and the corresponding SDE can be written as

$$d\boldsymbol{x}(t) = F(\boldsymbol{x}(t), \mathbf{W}(t))dt + \gamma \boldsymbol{x}(t) \odot d\mathbf{B}(t),\ \gamma > 0, \tag{7}$$

where $\odot$ denotes the Hadamard product. Similarly, we can use the forward and backward Euler-Maruyama discretizations of (7) to propagate the image in Fig. 1 (a), and we provide these results in Appendix D.1. The corresponding residual perturbation is

$$\boldsymbol{x}^{i+1} = \boldsymbol{x}^i + \hat{F}(\boldsymbol{x}^i, \mathbf{W}^i) + \gamma \boldsymbol{x}^i \odot \boldsymbol{n}^i,\ \text{where } \boldsymbol{n}^i \sim \mathcal{N}(\mathbf{0}, \mathbf{I}), \tag{8}$$

again, the noise $\gamma \boldsymbol{x}^i \odot \boldsymbol{n}^i$ is injected to each residual mapping in both training and test phases.

We will provide theoretical guarantees for these two residual perturbation schemes, i.e., (6) and (8), in Section 3, and numerically verify their efficacy in Section 4.

## 2.3 UTILITY ENHANCEMENT VIA MODEL ENSEMBLE

Wang et al. (2019b) showed that an ensemble of noise injected ResNets can improve models' utility. In this paper, we will also study the model ensemble for utility enhancement. We inherit notations from (Wang et al., 2019b), e.g., we denote an ensemble of two noise injected ResNet8 as En$_2$ResNet8.

## 3 MAIN THEORY

In this section, we will provide theoretical guarantees for the above two residual perturbations.

---

[1] Liu et al. (2019); Wang et al. (2019b) used this noise injection to improve robustness of ResNets.
[2] Liu et al. (2019) injected multiplicative noise to neural networks to improve their robustness.

## 3.1 Differential Privacy Guarantee for Strategy I

We consider the following function class for ResNets with residual perturbation:

$$\mathcal{F}_1 := \{f(\boldsymbol{x}) = \boldsymbol{w}^{\mathrm{T}}\boldsymbol{x}^M | \boldsymbol{x}^{i+1} = \boldsymbol{x}^i + \phi\left(\mathbf{U}^i\boldsymbol{x}^i\right) + \gamma\boldsymbol{n}^i, i = 0, \cdots, M-1, \quad (9)$$
$$\boldsymbol{x}^0 = \text{input data} + \pi\boldsymbol{n}, \boldsymbol{n} \text{ and } \boldsymbol{n}^i \sim \mathcal{N}(\mathbf{0}, \mathbf{I}), \boldsymbol{w} \in \mathbb{R}^d, \mathbf{U}^i \in \mathbb{R}^{d\times d}\},$$

where $\boldsymbol{x}^0 \in \mathbb{R}^d$ is the noisy input [3], $\mathbf{U}^i$ is the weight matrix in the $i$th residual mapping and $\boldsymbol{w} \in \mathbb{R}^d$ is the weights of the last layer. $\gamma, \pi > 0$ are hyperparameters. $\phi = \text{BN}(\psi)$ with BN being the batch normalization and $\psi$ being a $L$-Lipschitz and monotonically increasing activation function (e.g., ReLU). We first recap on the definition of differential privacy below.

**Definition 1** (($\epsilon, \delta$)-DP). *(Dwork et al., 2006)A randomized mechanism $\mathcal{M} : \mathcal{S}^N \to \mathcal{R}$ satisfies $(\epsilon, \delta)$-DP if for any two datasets $S, S' \in \mathcal{S}^N$ that differ by one element, and any output subset $O \subseteq \mathcal{R}$, it holds that $\mathbb{P}[\mathcal{M}(S) \in O] \le e^\epsilon \cdot \mathbb{P}[\mathcal{M}(S') \in O] + \delta,$ where $\delta \in (0, 1)$ and $\epsilon > 0$.*

We have the following DP guarantee for **Strategy I**, and we provide its proof in Appendix A.

**Theorem 1.** *Assume the input to ResNet lies in $B(\mathbf{0}, R)$ and the output of every residual mapping is normal distributed and bounded by $G$, in $\ell_2$ norm, in expectation. Given the total number of iterations $T$ used for training ResNet. For any $\epsilon > 0$ and $\delta, \lambda \in (0, 1)$, the parameters $\mathbf{U}^i$ and $\boldsymbol{w}$ in the ResNet with residual perturbation satisfies $((\lambda/i + (1 - \lambda))\epsilon, \delta)$-DP and $((\lambda/M + (1 - \lambda))\epsilon, \delta)$-DP, respectively, provided that $\pi > R\sqrt{(2Tb\alpha)/(N\lambda\epsilon)}$ and $\gamma > G\sqrt{(2Tb\alpha)/(N\lambda\epsilon)}$, where $\alpha = \log(1/\delta)/((1 - \lambda)\epsilon) + 1$, $M, N$ and $b$ are the number of residual mappings, training data, and batch size, respectively. In particular, when $\gamma > G\sqrt{(2Tb\alpha)/(NM\lambda\epsilon)}$ the whole model obtained by injecting noise according to strategy I satisfies $(\epsilon, \delta)$-DP.*

## 3.2 Theoretical Guarantees for Strategy II

**Privacy.** To analyze the residual perturbation (8), we consider the following function class:

$$\mathcal{F}_2 := \{f(\boldsymbol{x}) = \boldsymbol{w}^{\mathrm{T}}\boldsymbol{x}^M + \pi\boldsymbol{x}^M\boldsymbol{n} | \boldsymbol{x}^{i+1} = \boldsymbol{x}^i + \phi\left(\mathbf{U}^i\boldsymbol{x}^i\right) + \gamma\tilde{\boldsymbol{x}}^i \odot \boldsymbol{n}^i), \quad (10)$$
$$i = 0, \cdots, M-1, \boldsymbol{n}^i \sim \mathcal{N}(\mathbf{0}, \mathbf{I}), \|\boldsymbol{w}\|_2 \le a\}$$

where $a > 0$ is a constant; we denote the entry of $\boldsymbol{x}^i$ that has the largest absolute value as $x^i_{max}$, and $\tilde{\boldsymbol{x}}^i$ is defined as $(sgn(x^i_j)\max(|x^i_j|, \eta))^d_{j=1}$. Due to batch normalization, we assume $\phi$ can be bounded by a positive constant $B$. The other notations are defined similar to that in (9).

Consider training $\mathcal{F}_2$ by using two different datasets $S$ and $S'$, and we denote the resulting models as:

$$f(\boldsymbol{x}|S) := \boldsymbol{w}_1^{\mathrm{T}}\boldsymbol{x}^M + \pi\boldsymbol{x}^M\boldsymbol{n}^M; \ \boldsymbol{x}^{i+1} = \boldsymbol{x}^i + \phi\left(\mathbf{U}_1^i\boldsymbol{x}^i\right) + \gamma\tilde{\boldsymbol{x}}^i \odot \boldsymbol{n}^i, i = 0, \cdots, M-1. \quad (11)$$
$$f\left(\boldsymbol{x}|S'\right) := \boldsymbol{w}_2^{\mathrm{T}}\boldsymbol{x}^M + \pi\boldsymbol{x}^M\boldsymbol{n}^M; \ \boldsymbol{x}^{i+1} = \boldsymbol{x}^i + \phi\left(\mathbf{U}_2^i\boldsymbol{x}^i\right) + \gamma\tilde{\boldsymbol{x}}^i \odot \boldsymbol{n}^i, i = 0, \cdots, M-1. \quad (12)$$

**Theorem 2.** *For $f(\boldsymbol{x}|S)$ and $f(\boldsymbol{x}|S')$ that are defined in (11) and (12), respectively. Let $\lambda \in (0, 1), \delta \in (0, 1)$, and $\epsilon > 0$, if $\gamma > (B/\eta)\sqrt{(2\alpha M)/(\lambda\epsilon)}$ and $\pi > a\sqrt{(2\alpha M)/\lambda\epsilon}$, where $\alpha = \log(1/\delta)/((1 - \lambda)\epsilon) + 1$, then $\mathbb{P}[f(\boldsymbol{x}|S) \in O] \le e^\epsilon \cdot \mathbb{P}[f(\boldsymbol{x}|S') \in O] + \delta$ for any input $\boldsymbol{x}$ and any subset $O$ in the output space.*

We provide the proof of Theorem 2 in Appendix B. Theorem 2 guarantees the privacy of the training data given only black-box access to the model, i.e., the model will output the prediction for any input without granting adversaries access to the model itself. In particular, we cannot infer whether the model is trained on $S$ or $S'$ no matter how we query the model in a black-box fashion. We leave the theoretical DP-guarantee for for Strategy II as a future work.

**Generalization Gap.** Many works have shown that overfitting in training ML models leads to privacy leakage (Salem et al., 2018), and reducing overfitting can mitigate data privacy leakage (Shokri et al., 2017; Yeom et al., 2018; Sablayrolles et al., 2018; Salem et al., 2018; Wu et al., 2019b). In this part, we will show that the residual perturbation (8) can reduce overfitting via computing the Rademacher complexity. For simplicity, we consider binary classification problems. Suppose $S_N = \{\boldsymbol{x}_i, y_i\}^N_{i=1}$ is drawn from $X \times Y \subset \mathbb{R}^d \times \{-1, +1\}$ with $X$ and $Y$ being the input data and label

---

[3]We add noise to the input for DP guarantee

spaces, respectively. Assume $\mathcal{D}$ is the underlying distribution of $X \times Y$, which is unknown. Let $\mathcal{H} \subset V$ be the hypothesis class of the ML model. We first recap on the definition of Rademacher complexity.

**Definition 2.** *(Barlett & Mendelson, 2002) Let $\mathcal{H} : X \to \mathbb{R}$ be the space of real-valued functions on the space $X$. For a given sample $S = \{\boldsymbol{x}_1, \boldsymbol{x}_2, \cdots, \boldsymbol{x}_N\}$ of size $N$, the empirical Rademacher complexity of $\mathcal{H}$ is defined as*

$$R_S(\mathcal{H}) := \frac{1}{N} E_\sigma [\sup_{h \in \mathcal{H}} \sum_{i=1}^{N} \sigma_i h(\boldsymbol{x}_i)],$$

*where $\sigma_1, \sigma_2, \cdots, \sigma_N$ are i.i.d. Rademacher random variables with $\mathbb{P}(\sigma_i = 1) = \mathbb{P}(\sigma_i = -1) = \frac{1}{2}$.*

Rademacher complexity is a tool to bound the generalization gap (Barlett & Mendelson, 2002). The smaller the generalization gap is, the less overfitting the model is. For $\forall \boldsymbol{x}_i \in \mathbb{R}^d$ and constant $c \geq 0$, we consider the following two function classes:

$$\mathcal{F} := \{f(\boldsymbol{x}, \boldsymbol{w}) = \boldsymbol{w} \boldsymbol{x}^p(T) | d\boldsymbol{x}(t) = \mathbf{U} \boldsymbol{x}(t) dt, \boldsymbol{x}(0) = \boldsymbol{x}_i; \ \boldsymbol{w} \in \mathbb{R}^{1 \times d}, \mathbf{U} \in \mathbb{R}^{d \times d} \}$$
$$\text{with } \|\boldsymbol{w}\|_2, \|\mathbf{U}\|_2 \leq c\},$$

$$\mathcal{G} := \{f(\boldsymbol{x}, \boldsymbol{w}) = \mathbb{E}(\boldsymbol{w} \boldsymbol{x}^p(T)) | d\boldsymbol{x}(t) = \mathbf{U} \boldsymbol{x}(t) dt + \gamma \boldsymbol{x}(t) \odot dB(t), \boldsymbol{x}(0) = \boldsymbol{x}_i; \ \boldsymbol{w} \in \mathbb{R}^{1 \times d},$$
$$\text{with } \mathbf{U} \in \mathbb{R}^{d \times d}, \|\boldsymbol{w}\|_2, \|\mathbf{U}\|_2 \leq c\},$$

where $0 < p < 1$ takes the value such that $x^p$ is well defined on the whole $\mathbb{R}^d$. $\gamma > 0$ is a hyperparameter and $\mathbf{U}$ is a circulant matrix that corresponding to the convolution layer in DNs. $B(t)$ being the 1D Brownian motion. The function class $\mathcal{F}$ represents the continuous analogue of ResNet without inner nonlinear activation functions, and $\mathcal{G}$ denotes $\mathcal{F}$ with the residual perturbation (8).

**Theorem 3.** *Given the training set $S_N = \{\boldsymbol{x}_i, y_i\}_{i=1}^N$. We have $R_{S_N}(\mathcal{G}) < R_{S_N}(\mathcal{F})$.*

We provide the proof of Theorem 3 in Appendix C, where we will also provide quantitative lower and upper bounds of the above Rademacher complexities. Theorem 3 shows that residual perturbation (8) can reduce the generalization error. We will numerically verify this generalization error reduction for ResNet with residual perturbation in Section 4.

## 4 EXPERIMENTS

In this section, we will numerically verify that 1) can residual perturbation protect data privacy; in particular, membership privacy? 2) can the ensemble of ResNets with residual perturbation improve the classification accuracy? 3) are skip connections crucial in residual perturbation for DL with data privacy? 4) what is the advantage of the residual perturbation over the DPSGD? We focus on **Strategy I** in this section, and we provide the results of **Strategy II** in Appendix D.

### 4.1 PRELIMINARIES

**Datasets.** We consider both CIFAR10/CIFAR100 (Krizhevsky et al., 2009) and the Invasive Ductal Carcinoma (IDC) datasets. Both CIFAR10 and CIFAR100 contain 60K $32 \times 32$ color images with 50K and 10K of them used for training and test, respectively. The IDC dataset is a breast cancer-related benchmark dataset, which contains 277,524 patches of the size $50 \times 50$ with 198,738 labeled negative (0) and 78,786 labeled positive (1). Figure 2 depicts a few patches from the IDC dataset. For the IDC dataset, we follow Wu et al. (2019a) and split the whole dataset into training, validation, and test set. The training set consists of 10,788 positive patches and 29,164 negative patches, and the test set contains 11,595 positive patches and 31,825 negative patches. The remaining patches are used as the validation set. For each dataset, we split its training set into $D_{\text{shadow}}$ and $D_{\text{target}}$ with the same size. Furthermore, we split $D_{\text{shadow}}$ into two halves with the same size and denote them as $D_{\text{shadow}}^{\text{train}}$ and $D_{\text{shadow}}^{\text{out}}$, and split $D_{\text{target}}$ by half into $D_{\text{target}}^{\text{train}}$ and $D_{\text{target}}^{\text{out}}$. The purpose of this splitting of the training set is for the membership inference attack, which will be discussed below.

**Membership inference attack.** To verify the efficiency of residual perturbation for protecting data privacy, we consider the membership inference attack (Salem et al., 2018) in all the experiments below. The membership attack proceeds as follows: 1) train the shadow model by using $D_{\text{shadow}}^{\text{train}}$; 2) apply the trained shadow model to predict all data points in $D_{\text{shadow}}$ and obtain the corresponding

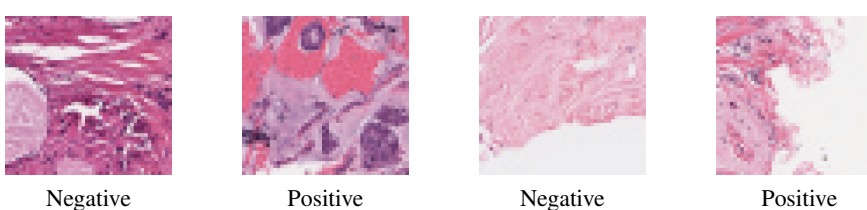

| Negative | Positive | Negative | Positive |

Figure 2: Visualization of a few selected images from the IDC dataset.

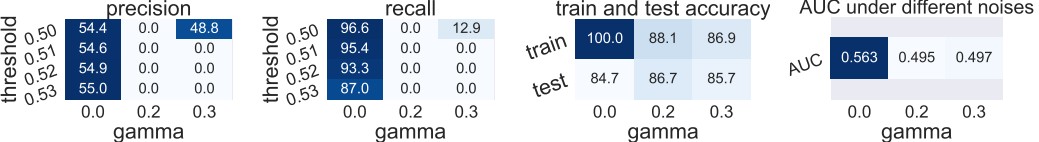

Figure 3: Performance of residual perturbation for $En_5ResNet8$ with different noise coefficients ($\gamma$) and membership inference attack thresholds on the IDC dataset. Residual perturbation can significantly improve membership privacy and reduce the generalization gap. $\gamma = 0$ corresponding to the baseline ResNet8. (Unit: %)

classification probabilities of belonging to each class. Then we take the top three classification probabilities (or two in the case of binary classification) to form the feature vector for each data point. A feature vector is tagged as 1 if the corresponding data point is in $D_{shadow}^{train}$, and 0 otherwise. Then we train the attack model by leveraging all the labeled feature vectors; 3) train the target model by using $D_{target}^{train}$ and obtain feature vector for each point in $D_{target}$. Finally, we leverage the attack model to decide whether a data point is in $D_{target}^{train}$.

**Experimental settings.** We consider $En_5ResNet8$ (ensemble of 5 ResNet8 with residual perturbation) and the standard ResNet8 as the target and shadow models. We use a multilayer perceptron with a hidden layer of 64 nodes, followed by a softmax output function as the attack model, which is adapted from (Salem et al., 2018). We apply the same settings as that used in (He et al., 2016) to train the target and shadow models on the CIFAR10 and CIFAR100. For training models on the IDC dataset, we run 100 epochs of SGD with the same setting as before except that we decay the learning rate by 4 at the 20th, 40th, 60th, and 80th epoch, respectively. Moreover, we run 50 epochs of Adam (Kingma & Ba, 2014) with a learning rate of 0.1 to train the attack model. For both IDC and CIFAR datasets, we set $\pi$ as half of $\gamma$ in **Strategy I**, which simplifies hyperparameters calibration, and based on our experiment it gives a good trade-off between privacy and accuracy.

**Performance evaluations.** We consider both classification accuracy and capability for protecting membership privacy. The attack model is a binary classifier, which is to decide if a data point is in the training set of the target model. For any $x \in D_{target}$, we apply the attack model to predict its probability ($p$) of belonging to the training set of the target model. Given any fixed threshold $t$ if $p \geq t$, we classify $x$ as a member of the training set (positive sample), and if $p < t$, we conclude that $x$ is not in the training set (negative sample); so we can obtain different attack results with different thresholds. Furthermore, we can plot the ROC curve(see details in subsection 4.5) of the attack model and use the area under the ROC curve (AUC) as an evaluation of the membership inference attack. The target model protects perfect membership privacy if the AUC is 0.5 (attack model performs random guess), and the higher AUC is, the less private the target model is. Moreover, we use the precision (the fraction of records inferred as members are indeed members of the training set) and recall (the fraction of training set that is correctly inferred as members of the training set by the attack model) to measure ResNets' capability for protecting membership privacy.

## 4.2 EXPERIMENTS ON THE IDC DATASET

In this subsection, we numerically verify that the residual perturbation in protecting data privacy while retaining the classification accuracy on the IDC dataset. We select the $En_5ResNet8$ as a benchmark architecture, which has ResNet8 as its baseline architecture (the details of the neural architectures are provided in Appendix F). As shown in Figure 3, we set four different thresholds to obtain different attack results with three different noise coefficients ($\gamma$) when $\gamma = 0$ means the standard ResNet8 without residual perturbation. We also depict the ROC curve for this experiment in Figure 7 (c).

Table 1: Residual perturbation vs. DPSGD in training ResNet8 and EnResNet8 for the IDC classification. Ensemble of ResNet8 with residual perturbation has higher test accuracy and protects better membership privacy (smaller AUC).

| | ResNet8 (DPSGD) | | | $En_1$ResNet8 | | | $En_5$ResNet8 | |
|---|---|---|---|---|---|---|---|---|
| AUC | Training Acc | Test Acc | AUC | Training Acc | Test Acc | AUC | Training Acc | Test Acc |
| 0.503 | 0.831 | 0.828 | 0.496 | 0.864 | 0.852 | 0.497 | 0.869 | 0.857 |

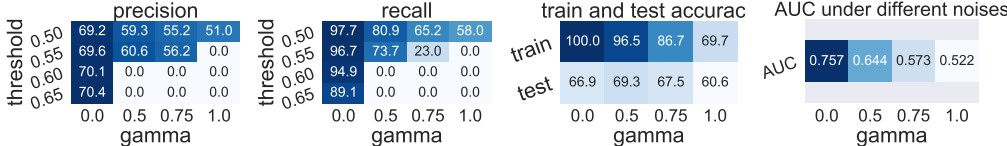

Figure 4: Performance of $En_5$ResNet8 with residual perturbation using different noise coefficients ($\gamma$) and membership inference attack threshold on CIFAR10. Residual perturbation can not only enhance the membership privacy, but also improve the classification accuracy. $\gamma = 0$ corresponding to the baseline ResNet8 without residual perturbation or model ensemble. (Unit: %)

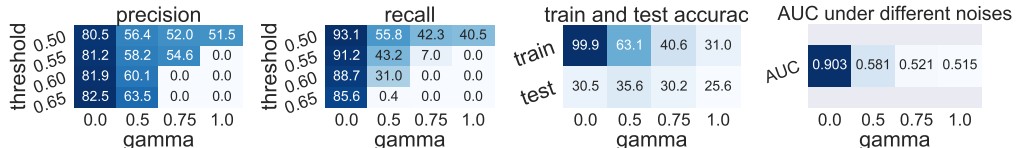

Figure 5: Performance of $En_5$ResNet8 with residual perturbation using different noise coefficients ($\gamma$) and membership inference attack threshold on CIFAR100. Again, residual perturbation can not only enhance the membership privacy, but also improve the classification accuracy. $\gamma = 0$ corresponding to the baseline ResNet8 without residual perturbation or model ensemble. (Unit: %)

$En_5$ResNet is remarkably better in protecting the membership privacy, and as $\gamma$ increases the model becomes more resistant to the membership attack. Also, as the noise coefficient increases, the gap between training and test accuracies becomes smaller, which resonates with Theorem 3. For instance, when $\gamma = 0.2$ the AUC for the attack model is 0.495 and 0.563, respectively, for $En_5$ResNet8 and ResNet8; the classification accuracy of $En_5$ResNet8 and ResNet8 are 0.867 and 0.847, respectively.

### 4.2.1 RESIDUAL PERTURBATION VS. DPSGD

In this part, we compare the residual perturbation with the benchmark Tensorflow DPSGD module (McMahan et al., 2018), and we calibrate the hyperparameters, including the initial learning rate (0.1) which decays by a factor of 4 after every 20 epochs, noise multiplier (1.1), clipping threshold (1.0), micro-batches (128), and epochs (100) [4] such that the resulting model gives the optimal trade-off between membership privacy and classification accuracy. DPSGD is significantly more expensive due to the requirement of computing the per-sample gradient. We compare the standard ResNet8 trained by DPSGD with $En_1$ResNet8 and $En_5$ResNet8 with residual perturbation ($\gamma = 0.3$). Table 1 lists the AUC of the attack model and training and test accuracies of the target model; we see that residual perturbation can improve accuracy and protect better membership privacy.

### 4.3 EXPERIMENTS ON THE CIFAR10/CIFAR100 DATASETS

We further test the residual perturbation for ResNet8 and $En_5$ResNet8 on the CIFAR10/CIFAR100 dataset. Figure 4 plots the performance of $En_5$ResNet8 on the CIFAR10 dataset under the above four different measures. Again, the ensemble of ResNets with residual perturbation is remarkably less vulnerable to the membership inference attack; for instance, the AUC of the attack model for ResNet8 and $En_5$ResNet8 ($\gamma = 0.75$) is 0.757 and 0.573, respectively. Also, the classification accuracy of $En_5$ResNet8 (67.5%) is higher than that of ResNet8 (66.9%) for CIFAR10 classification. Figure 5 depicts the results of $En_5$ResNet8 for CIFAR100 classification. These results confirm again that residual perturbation can protect membership privacy and improve classification accuracy.

---

[4]https://github.com/tensorflow/privacy/tree/master/tutorials

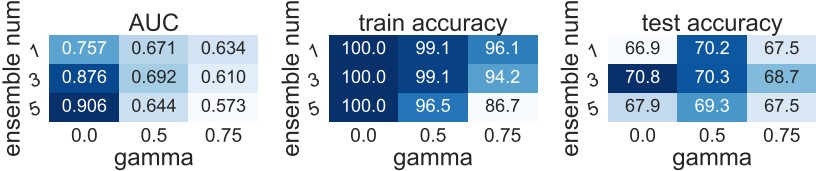

Figure 6: The performance of residual perturbation with different noise coefficients ($\gamma$) and different number of models in the ensemble. The optimal privacy-utility tradeoff lies in the choice of these two options. (Unit: %)

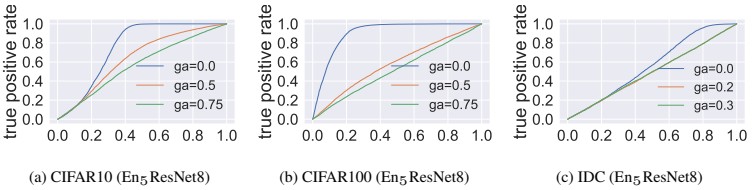

(a) CIFAR10 (En$_5$ResNet8)  (b) CIFAR100 (En$_5$ResNet8)  (c) IDC (En$_5$ResNet8)

Figure 7: ROC curves for different datasets. (ga: noise coefficient $\gamma$)

### 4.3.1 EFFECTS OF THE NUMBER OF MODELS IN THE ENSEMBLE

In this part, we consider the effects of the number of residual perturbed ResNets in the ensemble. Figure 6 illustrates the performance of EnResNet8 for CIFAR10 classification measured in the AUC, and training and test accuracy. These results show that tuning the noise coefficient and the number of models in the ensemble is crucial to optimize the trade-off between accuracy and privacy.

### 4.4 ON THE IMPORTANCE OF SKIP CONNECTIONS

Residual perturbations theoretically relies on the irreversibility of the SDEs (3) and (7), and this ansatz lies in the skip connections in the ResNet. We test both standard ResNet and the modified ResNet without skip connections. For CIFAR10 classification, under the same noise coefficient ($\gamma = 0.75$), the test accuracy is $0.675$ for the En$_5$ResNet8 (with skip connection); while the test accuracy is $0.653$ for the En$_5$ResNet8 (without skip connection). Skip connections makes EnResNet more resistant to noise injection which is indeed crucial for the success of residual perturbation for protecting data privacy.

### 4.5 ROC CURVES FOR EXPERIMENTS ON DIFFERENT DATASETS

The receiver operating characteristic (ROC) curve can be used to illustrate the classification ability of a binary classifier. ROC curve is obtained by plotting the true positive rate against the false positive rate at different thresholds. The true positive rate, also known as recall, is the fraction of the positive set (all the positive samples) that is correctly inferred as a positive sample by the binary classifier. The false positive rate can be calculated by 1-specificity, where specificity is the fraction of the negative set (all the negative samples) that is correctly inferred as a negative sample by the binary classifier. In our case, the attack model is a binary classifier. Data points in the training set of the target model are tagged as positive samples, and data points out of the training set of the target model are tagged as negative samples. Then we plot ROC curves for different datasets (as shown in Figure 7). These ROC curves show that if $\gamma$ is sufficiently large, the attack model's prediction will be nearly a random guess.

### 4.6 REMARK ON THE PRIVACY BUDGET

In the experiments above, we set the constants $G$ and $R$ to 30 for **Strategy I**. For classifying IDC with ResNet8, the DP budget for **Strategy I** is ($\epsilon = 1.1e5, \delta = 1e - 5$) and the DP-budget for DPSGD is ($\epsilon = 15.79, \delta = 1e - 5$). For classifying CIFAR10 with ResNet8, the DP budget for **Strategy I** is ($\epsilon = 3e5, \delta = 1e - 5$) and the DP-budget for DPSGD is ($\epsilon = 22.33, \delta = 1e - 5$). Note that theorem 1 offers a quite loose DP budget compared to DPSGD. There are several difficulties we need to overcome to get tight DP bounds for **Strategy I**. Compared to DPSGD, it is significantly harder. In particular, 1) the loss function of the nose injected ResNets is highly nonlinear and very complex with respect to the weights, also the noise term appears in the loss function due to the noise injected in each residual mapping. These together make the tight estimate very difficult. 2) In our proof, we leveraged the framework of subsampled Rényi-DP (Wang et al., 2018) to find a feasible range of

noise variance parameter, and then convert to DP to get the value of $\gamma$ for a given DP budget. This procedure will significantly reduce the accuracy of the estimated $\gamma$. We leave the tight DP guarantee as future work. In particular, how to reduce the accuracy of estimating due to the conversion between Rényi-DP and DP.

## 5 CONCLUDING REMARKS

In this paper, we proposed residual perturbations, whose theoretical foundation lies in the theory of stochastic differential equations, to protect data privacy for deep learning. Theoretically, we prove that the residual perturbation can reduce the generalization gap with differential privacy guarantees. Numerically, we have shown that residual perturbations are effective for protecting membership privacy on some benchmark datasets. In particular, on the IDC benchmark, residual perturbations protect better membership privacy than state-of-the-art differentially private stochastic gradient descent and achieve remarkably better classification accuracy.

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

# Part

# Appendices

## Table of Contents

## A  PROOF OF THEOREM 1

### A.1  RÉNYI DIFFERENTIAL PRIVACY

We will use the notion of Rényi differential privacy (RDP) to prove the differential privacy (DP) guarantees for the proposed residual perturbations. First, let's review the definition and several results of the Rényi differential privacy (Mironov, 2017).

**Definition 3.** *(Mironov, 2017) (Rényi divergence) For any two probability distributions $P$ and $Q$ defined over the distribution $\mathcal{D}$, the Rényi divergence of order $\alpha > 1$ is*

$$D_\alpha(P||Q) = \frac{1}{\alpha - 1} \log \mathbb{E}_{x \sim Q}(P/Q)^\alpha.$$

**Definition 4.** *(Mironov, 2017) ($(\alpha, \epsilon)$-RDP) A randomized mechanism $\mathcal{M} : \mathcal{D} \to \mathcal{R}$ is said to have $\epsilon$-Rényi differential privacy of order $\alpha$ or $(\alpha, \epsilon)$-RDP for short, if for any adjacent $\mathcal{S}, \mathcal{S}' \in \mathcal{D}$ that differ by only one entry, it holds that*

$$D_\alpha(\mathcal{M}(\mathcal{S})||\mathcal{M}(\mathcal{S}')) \le \varepsilon.$$

**Lemma 1.** *(Mironov, 2017) Let $f : \mathcal{D} \to \mathcal{R}_1$ be $(\alpha, \epsilon_1)$-RDP and $g : \mathcal{R}_1 \times \mathcal{D} \to \mathcal{R}$ be $(\alpha, \epsilon)$-RDP, then the mechanism defined as $(X, Y)$, where $X \sim f(D)$ and $Y \sim g(X, D)$, satisfies $(\alpha, \epsilon_1 + \epsilon_2)$-RDP.*

**Lemma 2.** *(Mironov, 2017) (From RDP to $(\epsilon, \delta)$-DP) If $f$ is an $(\alpha, \epsilon)$-RDP mechanism, then it also satisfies $(\epsilon - (\log \delta)/(\alpha - 1), \delta)$-differential privacy for any $0 < \delta < 1$.*

**Lemma 3.** *(Mironov, 2017) (Post-processing lemma) Let $\mathcal{M} : \mathcal{D} \to \mathcal{R}$ be a randomized algorithm that is $(\alpha, \epsilon)$-RDP, and let $f : \mathcal{R} \to \mathcal{R}'$ be an arbitrary randomized mapping. Then $f(\mathcal{M}(\cdot)) : \mathcal{D} \to \mathcal{R}'$ is $(\alpha, \epsilon)$-RDP*

### A.2 PROOF OF THEOREM 1

In this subsection, we will give a proof of Theorem 1, i.e., DP-guarantee for the **Strategy I**.

*Proof.* We will prove Theorem 1 by mathematical induction. Consider two adjacent datasets $\mathcal{S} = \{\boldsymbol{x}_1, \cdots, \boldsymbol{x}_{N-1}, \boldsymbol{x}_N\}, \mathcal{S}' = \{\boldsymbol{x}_1, \cdots, \boldsymbol{x}_{N-1}, \hat{\boldsymbol{x}}_N\}$ that differ by one entry. For the first residual mapping, it is easy to check that when $\gamma \geq R\sqrt{2\alpha/\epsilon_p}$ we have $D_\alpha(\boldsymbol{x}_N^1 || \hat{\boldsymbol{x}}_N^1) = \alpha \|\boldsymbol{x}_N - \hat{\boldsymbol{x}}_N\|^2/(2\gamma^2) < \varepsilon_p$. For the remaining residual mappings, we denote the response of the $i$th residual mapping, for any two input data $\boldsymbol{x}_N$ and $\hat{\boldsymbol{x}}_N$, as $\boldsymbol{x}_N^i, \hat{\boldsymbol{x}}_N^i$, respectively. Based on our assumption, we have

$$\boldsymbol{x}_N^i + \phi(\mathbf{U}^i \boldsymbol{x}_N^i) \sim \mathcal{N}(\mu_{N,i}, \sigma_{N,i}^2)$$
$$\hat{\boldsymbol{x}}_N^i + \phi(\mathbf{U}^i \hat{\boldsymbol{x}}_N^i) \sim \mathcal{N}(\hat{\mu}_{N,i}, \sigma_{N,i}^2),$$

where $\mu_{N,i}$ and $\hat{\mu}_{N,i}$ are both bounded by the constant $G$. If $D_\alpha(\boldsymbol{x}_N^i || \hat{\boldsymbol{x}}_N^i) \leq \epsilon_p/i$, according the post-processing lemma (Lemma 3), we have

$$D_\alpha(\boldsymbol{x}_N^i + \phi(\mathbf{U}^i \boldsymbol{x}_N^i) || \hat{\boldsymbol{x}}_N^i + \phi(\mathbf{U}^i \hat{\boldsymbol{x}}_N^i)) = \frac{\alpha \|\mu_{N,i} - \mu_{N,i}'\|^2}{2\sigma_{N,i}^2} \leq \epsilon_p/i,$$

so when $\gamma > \sqrt{2\alpha G^2/\epsilon_p}$, we further have

$$D_\alpha(\boldsymbol{x}_N^i + \phi(\mathbf{U}^i \boldsymbol{x}_N^i) + \gamma\boldsymbol{n} || \hat{\boldsymbol{x}}_N^i + \phi(\mathbf{U}^i \hat{\boldsymbol{x}}_N^i) + \gamma\boldsymbol{n}) = \frac{\alpha \|\mu_{N,i} - \mu_{N,i}'\|^2}{2(\sigma_{N,i}^2 + \gamma^2)} \leq \frac{\epsilon_p}{i+1},$$

which implies that $D_\alpha(\boldsymbol{x}_N^{i+1} || \hat{\boldsymbol{x}}_N^{i+1}) < \frac{\epsilon_p}{i+1}$. On the other hand, note that

$$\nabla_{\mathbf{U}^{i+1}} \ell |_{\boldsymbol{x}_N^{i+1}} = l'\left(\boldsymbol{x}_N^{i+1} + \phi\left(\mathbf{U}^{i+1} \boldsymbol{x}_N^{i+1}\right)\right) \phi'(\mathbf{U}^{i+1} \boldsymbol{x}_N^{i+1})(\boldsymbol{x}_N^{i+1})^T$$
$$\nabla_{\mathbf{U}^{i+1}} \ell |_{\hat{\boldsymbol{x}}_N^{i+1}} = l'\left(\hat{\boldsymbol{x}}_N^{i+1} + \phi\left(\mathbf{U}^{i+1} \hat{\boldsymbol{x}}_N^{i+1}\right)\right) \phi'(\mathbf{U}^{i+1} \hat{\boldsymbol{x}}_N^{i+1})(\hat{\boldsymbol{x}}_N^{i+1})^T.$$

Leveraging the post-processing lemma (Lemma 3) again, we get

$$D_\alpha(\nabla_{\mathbf{U}^{i+1}} \ell |_{\boldsymbol{x}_N^{i+1}} || \nabla_{\mathbf{U}^{i+1}} \ell |_{\hat{\boldsymbol{x}}_N^{i+1}}) < \frac{\epsilon_p}{i+1}.$$

Let $\mathcal{B}_t$ be the index set with $|\mathcal{B}_t| = b$, and we update $\mathbf{U}^i$ as following:

$$\mathbf{U}_{t+1}^{i+1} | \mathcal{S} = \mathbf{U}_t^i - \alpha \frac{1}{b} \sum_{j \in \mathcal{B}_t} \nabla_{\mathbf{U}_t^{i+1}} \ell(\boldsymbol{x}_j^{i+1}, \mathbf{U}_t^{i+1})$$

$$\mathbf{U}_{t+1}^{i+1} | \mathcal{S}' = \mathbf{U}_t^i - \alpha \frac{1}{b} \sum_{j \in \mathcal{B}_t} \nabla_{\mathbf{U}_t^{i+1}} \ell(\boldsymbol{x}_j^{i+1}, \mathbf{U}_t^{i+1}),$$

where $\mathbf{U}_t^{i+1}$ is the weights updated after the $t$th training iterations. When $N \notin \mathcal{B}_t$, it's obviously that $D_\alpha(\mathbf{U}_{t+1}^{i+1} | \mathcal{S} || \mathbf{U}_{t+1}^{i+1} | \mathcal{S}') = 0$; when $N \in \mathcal{B}_t$, the equations which we use to update $\mathbf{U}^i$ can be rewritten as

$$\mathbf{U}_{t+1}^{i+1} | \mathcal{S} = \mathbf{U}_t^i - \alpha(1/b) \sum_{j \in \mathcal{B}_t - \{N\}} \nabla_{\mathbf{U}_t^{i+1}} \ell(\boldsymbol{x}_j^{i+1}, \mathbf{U}_t^{i+1}) - \alpha(1/b) \nabla_{\mathbf{U}_t^{i+1}} \ell(\boldsymbol{x}_N^{i+1}, \mathbf{U}_t^{i+1})$$

$$\mathbf{U}_{t+1}^{i+1} | \mathcal{S}' = \mathbf{U}_t^i - \alpha(1/b) \sum_{j \in \mathcal{B}_t - \{N\}} \nabla_{\mathbf{U}_t^{i+1}} \ell(\boldsymbol{x}_j^{i+1}, \mathbf{U}_t^{i+1}) - \alpha(1/b) \nabla_{\mathbf{U}_t^{i+1}} \ell(\hat{\boldsymbol{x}}_N^{i+1}, \mathbf{U}_t^{i+1}).$$

According to the post-processing lemma (Lemma 3), we have $D_\alpha(\mathbf{U}_{t+1}^{i+1} | \mathcal{S} || \mathbf{U}_{t+1}^{i+1} | \mathcal{S}') \leq \epsilon_p/(i+1)$. Because there are only $(Tb)/N$ steps where we use the information of $\boldsymbol{x}_N$ and $\hat{\boldsymbol{x}}_N$. Replace $\epsilon_p$ by $(Tb\epsilon_p)/N$ and use composition theorem we can get after $T$ steps the output $\mathbf{U}_T^{i+1}$ satisfies $(\alpha, \epsilon_p/(i+1))$-RDP and $\boldsymbol{w}$ satisfies $(\alpha, \epsilon_p/M)$-RDP. By Lemma 2, we can easily establish the DP-guarantee for **Strategy I**, as stated in Theorem 1. $\square$

## B    PROOF OF THEOREM 2

In this section, we will provide a proof for Theorem 2.

*Proof.* Let $\phi = BN(\psi)$, where BN is batch normalization operation and $\psi$ is an activation function. Because of the property of batch normalization, we assume that $\phi$ can be bounded by a positive constant $B$. To show that the model (9) guarantees training data privacy given only black-box access to the model. Consider training the model (9) with two different datasets $S$ and $S'$, and we denote the resulting model as $f(\cdot|S)$ and $f(\cdot|S')$, respectively. In the following, we prove that with appropriate choices of $\gamma$ and $\pi$, $D_\alpha(f(\boldsymbol{x}|S)\|f(\boldsymbol{x}|S')) < \epsilon_p$ for any input $\boldsymbol{x}$.

First, we consider the convolution layers, and let $\text{Conv}(\boldsymbol{x})_i$ be the $i$th entry of the vectorized $\text{Conv}(\boldsymbol{x})$. Then we have $\text{Conv}(\boldsymbol{x})_i = \boldsymbol{x}_i + \phi(\mathbf{U}\boldsymbol{x})_i + \gamma\tilde{\boldsymbol{x}}_i\boldsymbol{n}_i$. For any two different training datasets, we denote

$$\text{Conv}(\boldsymbol{x})_i|S = \boldsymbol{x}_i + \phi(\mathbf{U}_1\boldsymbol{x})_i + \gamma\tilde{\boldsymbol{x}}_i\boldsymbol{n}_i \sim \mathcal{N}(\boldsymbol{x}_i + \phi(\mathbf{U}_1\boldsymbol{x})_i, \gamma^2\tilde{\boldsymbol{x}}_i^2),$$

and

$$\text{Conv}(\boldsymbol{x})_i|S' = \boldsymbol{x}_i + \phi(\mathbf{U}_2\boldsymbol{x})_i + \gamma\tilde{\boldsymbol{x}}_i\boldsymbol{n}_i \sim \mathcal{N}(\boldsymbol{x}_i + \phi(\mathbf{U}_2\boldsymbol{x})_i, \gamma^2\tilde{\boldsymbol{x}}_i^2).$$

Therefore, if $\gamma > (B/\eta)\sqrt{(2\alpha M)/(\epsilon_p)}$, we have

$$D_\alpha\left(\mathcal{N}\left(\boldsymbol{x}_i + \phi\left(\mathbf{U}_1\boldsymbol{x}\right)_i, \gamma^2\tilde{\boldsymbol{x}}_i^2\right)\|\mathcal{N}\left(\boldsymbol{x}_i + \phi\left(\mathbf{U}_2\boldsymbol{x}\right)_i, \gamma^2\tilde{\boldsymbol{x}}_i^2\right)\right)$$
$$\leq \frac{\alpha(\phi(\mathbf{U}_1\boldsymbol{x})_i - \phi(\mathbf{U}_2\boldsymbol{x})_i)^2}{2\gamma^2\eta^2} \leq \frac{4\alpha L^2 B^2\|\boldsymbol{x}\|_2^2}{2\gamma^2\tilde{\boldsymbol{x}}_i^2}$$
$$\leq \frac{2\alpha B^2}{\gamma^2\eta^2} \leq \epsilon_p/M.$$

Furthermore, $\gamma > (b/\eta)\sqrt{(2\alpha M)/(\epsilon_p)}$ guarantees $(\alpha, \epsilon_p/M)$-RDP for every convolution layer. For the last fully connected layer, if $\pi > a\sqrt{(2\alpha M)/\epsilon_p}$, we have

$$D_\alpha\left(\mathcal{N}\left(\boldsymbol{w}_1^T\boldsymbol{x}, \pi^2\tilde{\boldsymbol{x}}_i^2\right)\|\mathcal{N}\left(\boldsymbol{w}_2^T\boldsymbol{x}, \pi^2\tilde{\boldsymbol{x}}_i^2\right)\right)$$
$$= \frac{\alpha\left(\boldsymbol{w}_1^T\boldsymbol{x} - \boldsymbol{w}_2^T\boldsymbol{x}\right)^2}{2\pi^2\|\boldsymbol{x}\|_2} \leq \frac{4\alpha a^2\|\boldsymbol{x}\|_2^2}{2\pi^2\|\boldsymbol{x}\|_2}$$
$$\leq \frac{2\alpha a^2}{\pi^2} \leq \epsilon_p/M,$$

i.e., $\pi > a\sqrt{(2\alpha M)/\epsilon_p}$ guarantees that the fully connected layer to be $(\alpha, \epsilon_p/M)$-RDP. According to Lemma 1, we have $(\alpha, \epsilon_p)$-RDP guarantee for the ResNet of $M$ residual mappings if $\gamma > (Lb/\eta)\sqrt{(2\alpha d M)/(\epsilon_p)}$ and $\pi > a\sqrt{(2\alpha M)/\epsilon_p}$. Let $\lambda \in (0,1)$, for any given $(\epsilon_p, \delta)$ pair, if $\epsilon_p \leq \lambda\epsilon$ and $-(\log\delta)/(\alpha - 1), \delta) \leq (1 - \lambda)\epsilon$, then we have get the $(\epsilon, \delta)$-DP guarantee for the ResNet with $M$ residual mapping using the residual perturbation **Strategy II**.   □

## C    PROOF OF THEOREM 3

In this section, we will proof that the residual perturbation (8) can reduce the generalization error via computing the Rademacher complexity. Let us first recap on some related lemmas on the stochastic differential equation and the Rademacher complexity.

### C.1    SOME LEMMAS

Let $\ell : V \times Y \to [0, B]$ be the loss function. Here we assume $\ell$ is bounded and $B$ is a positive constant. In addition, we denote the function class $\ell_{\mathcal{H}} = \{(\boldsymbol{x}, y) \to \ell(h(\boldsymbol{x}), y) : h \in \mathcal{H}, (\boldsymbol{x}, y) \in X \times Y\}$. The goal of the learning problem is to find $h \in \mathcal{H}$ such that the population risk $R(h) = \mathbb{E}_{(\boldsymbol{x},y)\in\mathcal{D}}[\ell(h(\boldsymbol{x}), y)]$ is minimized. The gap between population risk and empirical risk $R_{S_N}(h) = (1/N)\sum_{i=1}^N \ell(h(\boldsymbol{x}_i), y_i)$ is known as the generalization error. We have the following lemma and theorem to connect the population and empirical risks via Rademacher complexity.

**Lemma 4.** *(Ledoux-Talagrand inequality) (M & Talagrand, 2002)Let $\mathcal{H}$ be a bounded real valued function space and let $\phi : \mathbb{R} \to \mathbb{R}$ be a Lipschitz with constant L and $\phi(0) = 0$. Then we have*

$$\frac{1}{n} E_\sigma \left[ \sup_{h \in \mathcal{H}} \sum_{i=1}^n \sigma_i \phi\left(h\left(x_i\right)\right) \right] \le \frac{L}{n} E_\sigma \left[ \sup_{h \in \mathcal{H}} \sum_{i=1}^n \sigma_i h\left(x_i\right) \right].$$

**Lemma 5.** *(Barlett & Mendelson, 2002)Let $S_N = \{(\boldsymbol{x}_1, y_1), \cdots, (\boldsymbol{x}_N, y_N)\}$ be samples chosen i.i.d. according to the distribution $\mathcal{D}$. If the loss function $\ell$ is bounded by $B > 0$. Then for any $\delta \in (0, 1)$, with probability at least $1 - \delta$, the following holds for all $h \in \mathcal{H}$,*

$$R\left(h\right) \le R_{S_N}\left(h\right) + 2B R_{S_N}\left(\ell_\mathcal{H}\right) + 3B\sqrt{(\log(2/\delta)/(2N)}.$$

*In addition, according to the Ledoux-Talagrand inequality and assume loss function is L-lipschitz, we have*

$$R_{S_N}\left(\ell_\mathcal{H}\right) \le L R_{S_N}\left(\mathcal{H}\right).$$

So the population risk can be bounded by the empirical risk and Rademacher complexity of the function class $\mathcal{H}$. Because we can't minimize the population risk directly, we can minimize it indirectly by minimizing the empirical risk and Rademacher complexity of the function class $\mathcal{H}$. Next, we will further discuss Rademacher complexity of the function class $\mathcal{H}$. We first introduce several lemmas below.

**Lemma 6.** *(Klebaner, 2005) For any given matrix $\mathbf{U}$, the solution to the equation*

$$\begin{cases} d\boldsymbol{x}(t) = \mathbf{U}\boldsymbol{x}(t)dt \\ \boldsymbol{x}(0) = \hat{\boldsymbol{x}} \end{cases}$$

*has the following expression*

$$\boldsymbol{x}(t) = \exp(\mathbf{U}t)\hat{\boldsymbol{x}},$$

*Also, we can write the solution to the following equation*

$$\begin{cases} d\boldsymbol{y}(t) = \mathbf{U}\boldsymbol{y}(t)dt + \gamma\boldsymbol{y}(t)dB(t) \\ \boldsymbol{y}(0) = \hat{\boldsymbol{x}} \end{cases}$$

*as*

$$\boldsymbol{y}(t) = \exp(\mathbf{U}t - \frac{1}{2}\gamma^2 t\mathbf{I} + \gamma B(t)\mathbf{I})\hat{\boldsymbol{x}}.$$

*Obviously, we have $\mathbb{E}[\boldsymbol{y}(t)] = \boldsymbol{x}(t)$.*

**Lemma 7.** *A matrix $\mathbf{C} \in \mathbb{R}^{d \times d}$ is circulant, if there exists real number $a_1, \cdots, a_d$ such that*

$$\mathbf{C} = \begin{pmatrix} a_1 & a_2 & \ddots & a_d \\ a_d & a_1 & \ddots & \ddots \\ \ddots & \ddots & \ddots & a_2 \\ a_2 & \ddots & a_d & a_1 \end{pmatrix}.$$

*For any circulant matrix, we have the following eigen-decomposition*

$$\Psi^H \mathbf{C} \Psi = diag(\lambda_1, \ldots, \lambda_d),$$

*where*

$$\sqrt{d}\Psi = \begin{pmatrix} 1 & 1 & \ldots & 1 \\ 1 & m_1 & \ldots & m_{d-1} \\ \ldots & \ldots & \ldots & \ldots \\ 1 & m_1^{d-1} & \ldots & m_{d-1}^{d-1} \end{pmatrix},$$

*and $m_i$s are the roots of unity and $\lambda_i = a_1 + a_2 m_i + \cdots + a_d m_i^{d-1}$.*

## C.2 THE PROOF OF THEOREM 3

*Proof.* By the definition of Rademacher complexity (Def. 2), we have

$$R_{S_N}(\mathcal{F}) = (1/N)\,\mathbb{E}_\sigma\left[\sup_{f\in\mathcal{F}}\sum_{i=1}^{N}\sigma_i \boldsymbol{w}\boldsymbol{x}_i^p(T)\right] = (c/N)\,\mathbb{E}_\sigma\left[\sup_{\|U\|_2\le c}\|\sum_{i=1}^{N}\sigma_i\boldsymbol{x}_i^p(T)\|_2\right].$$

Let $\boldsymbol{u}_i = \Psi\boldsymbol{x}_i^p$ and denote the $j$th element of $\boldsymbol{u}_i$ as $u_{i,j}$. Then by lemma 7, we have

$$R_{S_N}(\mathcal{F}) = (c/N)\,\mathbb{E}_\sigma \sup_{\|\mathbf{U}\|_2\le c}\left(\sum_{i,j}\sigma_i\sigma_j\langle\boldsymbol{x}_i^p(T),\boldsymbol{x}_j^p(T)\rangle\right)^{1/2}$$

$$= (c/N)\,\mathbb{E}_\sigma \sup_{|\lambda_i|\le c}(\|\Psi^H\sum_{i=1}^{N}\sigma_i\left(u_{i,1}\exp(\lambda_1 Tp),\cdots,u_{i,d}\exp(\lambda_d Tp)\right)^T\|_2)$$

$$= (c/N)\,\mathbb{E}_\sigma \sup_{|\lambda_i|\le c}\{\sum_{j=1}^{d}\left[\sum_{i=1}^{N}\sigma_i u_{i,j}\exp\left(\lambda_j Tp\right)\right]^2\}^{1/2} = (c/N)\exp(cTp)\mathbb{E}_\sigma\{\sum_{j=1}^{d}\left[\sum_{i=1}^{N}\sigma_i u_{i,j}\right]^2\}^{1/2}$$

$$= (c/N)\exp\left(cTp\right)\mathbb{E}_\sigma\|\sum_{i=1}^{N}\sigma_i\boldsymbol{u}_i\|_2 = (c/N)\exp\left(cTp\right)\mathbb{E}_\sigma\|\Psi\sum_{i=1}^{N}\sigma_i\boldsymbol{x}_i^p\|_2$$

$$= (c/N)\exp\left(cTp\right)\mathbb{E}_\sigma\|\sum_{i=1}^{N}\sigma_i\boldsymbol{x}_i^p\|_2$$

Note that $\mathbb{E}\left(\boldsymbol{w}\boldsymbol{x}_i^p(T)\right) = \boldsymbol{w}\mathbb{E}\left(\boldsymbol{x}_i^p(T)\right)$ and according to Lemma 6, similar to proof for the function class $\mathcal{F}$ we have

$$R_{S_N}(\mathcal{G}) = (c/N)\,\mathbb{E}_\sigma \sup_{\|\mathbf{U}\|_2\le c}\left(\sum_{i,j}\sigma_i\sigma_j\langle\mathbb{E}\boldsymbol{x}_i^p(T),\mathbb{E}\boldsymbol{x}_j^p(T)\rangle\right)^{1/2}$$

$$= (c/N)\mathbb{E}_\sigma \sup_{|\lambda_i|\le c}(\|\Psi^H\sum_{i=1}^{N}\sigma_i(u_{i,1}\exp(\lambda_1 Tp - p(1-p)\gamma^2 T/2),$$

$$\cdots, u_{i,d}\exp(\lambda_d Tp - p(1-p)\gamma^2 T/2))^T\|_2)$$

$$= (c/N)\,\mathbb{E}_\sigma \sup_{|\lambda_i|\le c}\{\sum_{j=1}^{d}\left[\sum_{i=1}^{N}\sigma_i u_{i,j}\exp\left(\lambda_j Tp - p(1-p)\gamma^2 T/2\right)\right]^2\}^{1/2}$$

$$= (c/N)\exp(cTp - p(1-p)\gamma^2 T/2)\mathbb{E}_\sigma\{\sum_{j=1}^{d}\left[\sum_{i=1}^{N}\sigma_i u_{i,j}\right]^2\}^{1/2}$$

$$= (c/N)\exp\left(cTp - p(1-p)\gamma^2 T/2\right)\mathbb{E}_\sigma\|\sum_{i=1}^{N}\sigma_i\boldsymbol{u}_i\|_2$$

$$= (c/N)\exp\left(cTp - p(1-p)\gamma^2 T/2\right)\mathbb{E}_\sigma\|\Psi\sum_{i=1}^{N}\sigma_i\boldsymbol{x}_i^p\|_2$$

$$= (c/N)\exp\left(cTp - p(1-p)\gamma^2 T/2\right)\mathbb{E}_\sigma\|\sum_{i=1}^{N}\sigma_i\boldsymbol{x}_i^p\|_2 < R_{S_N}(\mathcal{F})$$

Therefore, we have completed the proof for the fact that the ensemble of Gaussian noise injected ResNets can reduce generalization error compared to the standard ResNet. $\square$

# D  EXPERIMENTS ON STRATEGY II

## D.1  FORWARD AND BACKWARD PROPAGATION USING (7)

Figure 8 plots the forward and backward propagation of the ICLR logo using the SDE model (7). Again, we cannot simply use the backward Euler-Maruyama discretization to reverse the features generated by the propagating through the forward Euler-Maruyama discretization.

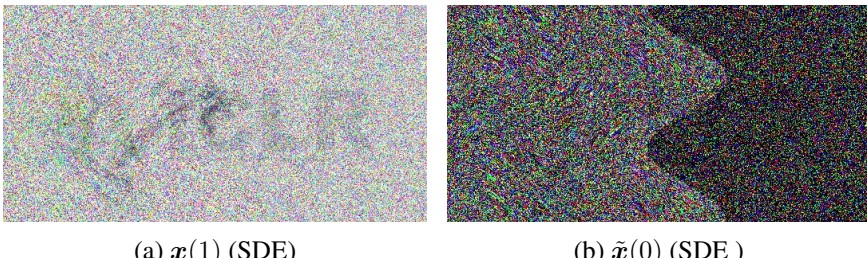

(a) $\boldsymbol{x}(1)$ (SDE)  (b) $\tilde{\boldsymbol{x}}(0)$ (SDE )

Figure 8: Illustrations of the forward and backward propagation of the training data using the SDE model (7). (a) is the features of the original image generated by the forward propagation using SDE; (b) is the recovered images by reverse-engineering the features shown in (a).

## D.2  EXPERIMENTS ON THE IDC DATASET

In this subsection, we consider the performance of the second residual perturbation in protecting membership privacy while retaining the classification accuracy on the IDC dataset. We use the same ResNet models as that used for the first residual perturbation. We list the results in Fig. 9, these results confirm that the residual perturbation (8) can effectively protect data privacy and maintain or even improve the classification accuracy. In addition, we depict the ROC curve for this experiment in Figure 12 (c). We note that, as the noise coefficient increases, the gap between training and testing accuracies narrows, which is consistent with Theorem 3.

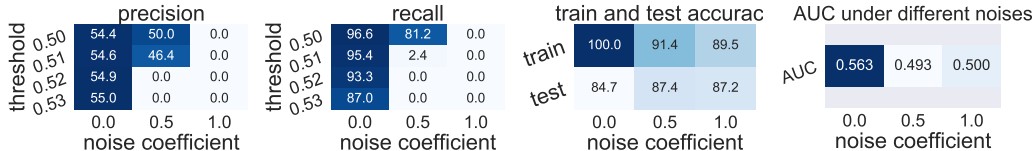

Figure 9: Performance of residual perturbation (8) for En$_5$ResNet8 with different noise coefficients ($\gamma$) and membership inference attack thresholds on the IDC dataset. Residual perturbation can significantly improve membership privacy and reduce the generalization gap. $\gamma = 0$ corresponding to the baseline ResNet8. (Unit: %)

### D.2.1  RESIDUAL PERTURBATION VS. DIFFERENTIALLY PRIVATE STOCHASTIC GRADIENT DESCENT

We have shown that the first residual perturbation (6) outperforms the DPSGD in protecting membership privacy and improving classification accuracy. In this part, we further show that the second residual perturbation (8) also outperforms the benchmark DPSGD with the above settings. Table 2 lists the AUC of the attack model and training & test accuracy of the target model; we see that the second residual perturbation can also improve the classification accuracy and protecting better membership privacy.

## D.3  EXPERIMENTS ON THE CIFAR10/CIFAR100 DATASETS

In this subsection, we will test the second residual perturbation (8) on the CIFAR10/CIFAR100 datasets with the same model using the same settings as before. Figure 10 plots the performance of En$_5$ResNet8 on the CIFAR10 dataset. These results show that the ensemble of ResNets with residual

Table 2: Residual perturbation (8) vs. DPSGD in training ResNet8 for the IDC dataset classification. Ensemble of ResNet8 with residual perturbation is more accurate for classification (higher test acc) and protects better membership privacy (smaller AUC).

| ResNet8 (DPSGD) | | | En$_1$ResNet8 | | | En$_5$ResNet8 | | |
|---|---|---|---|---|---|---|---|---|
| AUC | Training Acc | Test Acc | AUC | Training Acc | Test Acc | AUC | Training Acc | Test Acc |
| 0.503 | 0.831 | 0.828 | 0.509 | 0.880 | 0.868 | 0.500 | 0.895 | 0.872 |

perturbation (8) is significantly more robust to the membership inference attack. For instance, the AUC of the attack model for ResNet8 and En$_5$ResNet8 ($\gamma = 2.0$) is 0.757 and 0.526, respectively. Also, the classification accuracy of En$_5$ResNet8 ($\gamma = 2.0$) is higher than that of ResNet8, and their accuracy is 71.2% and 66.9% for CIFAR10 classification. Figure 11 shows the results of En$_5$ResNet8 for CIFAR100 classification. These results confirm that residual perturbation (8) can protect membership privacy and improve classification accuracy once more.

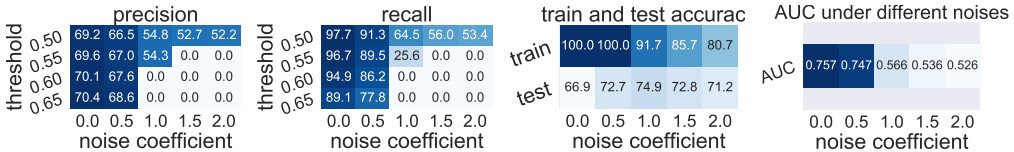

Figure 10: Performance of En$_5$ResNet8 with residual perturbation (8) using different noise coefficients ($\gamma$) and membership inference attack threshold on CIFAR10. Residual perturbation (8) can not only enhance the membership privacy, but also improve the classification accuracy. $\gamma = 0$ corresponding to the baseline ResNet8 without residual perturbation or model ensemble. (Unit: %)

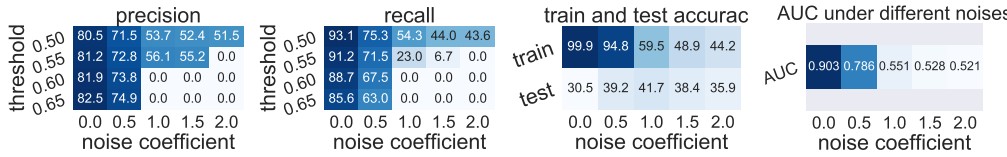

Figure 11: Performance of En$_5$ResNet8 with residual perturbation (8) using different noise coefficients ($\gamma$) and membership inference attack threshold on CIFAR100. Again, residual perturbation (8) can not only enhance the membership privacy, but also improve the classification accuracy. $\gamma = 0$ corresponding to the baseline ResNet8 without residual perturbation or model ensemble. (Unit: %)

### D.4    ROC CURVES FOR THE EXPERIMENTS ON DIFFERENT DATASETS

Figure 12 plots the ROC curves for the experiments on the different datasets with different models using the second residual perturbation strategy. These ROC curves again show that if $\gamma$ is sufficiently large, the attack model's prediction will be nearly a random guess.

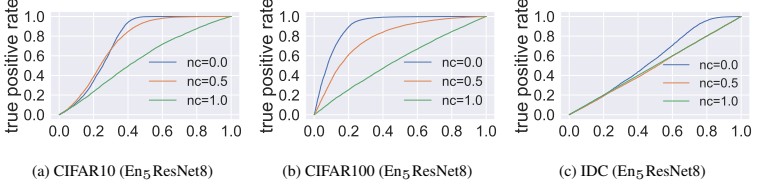

(a) CIFAR10 (En$_5$ResNet8)     (b) CIFAR100 (En$_5$ResNet8)     (c) IDC (En$_5$ResNet8)

Figure 12: ROC curves for different datasets. (nc: noise coefficient)

## E    MORE EXPERIMENTAL DETAILS

We give the detailed construction of the velocity field $F(\boldsymbol{x}(t), \mathbf{W}(t))$ in (3) and (7) that used to generate Figs. 1 and 8 in Algorithm 1.

---

**Algorithm 1** The expression of $F(\boldsymbol{x}(t), \mathbf{W}(t))$

---

**Input:** image=$\boldsymbol{x}(t)$; rows, cols, channels = image.shape.
**Output:** $F(\boldsymbol{x}(t), \mathbf{W}(t))$=dirtyimage.
**for** $k$ in range(channels) **do**
    **for** $i$ in range(rows): **do**
        **for** $j$ in range(cols): **do**
            offset($j$) =int($[j + 50.0\cos(2\pi i/180)]$%cols);
            offset(i) =int($[i + 50.0\sin(2\pi i/180)]$%row);
            dirtyimage$[i, j, k]$ = image$[(i + \text{offset}(j))$%rows$, (j + \text{offset}(i))$%cols$, k]$;
**return** dirtyimage

---

# F ARCHITECTURES OF THE USED DNS

Figure 13 shows the architectures of ResNets used in this paper, and we plot basic blocks in Fig. 14.

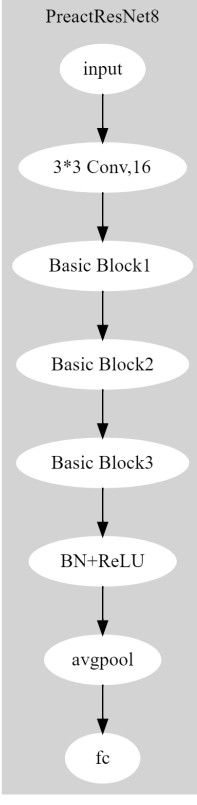

Figure 13: Architectures of the ResNet8 used in our experiments.

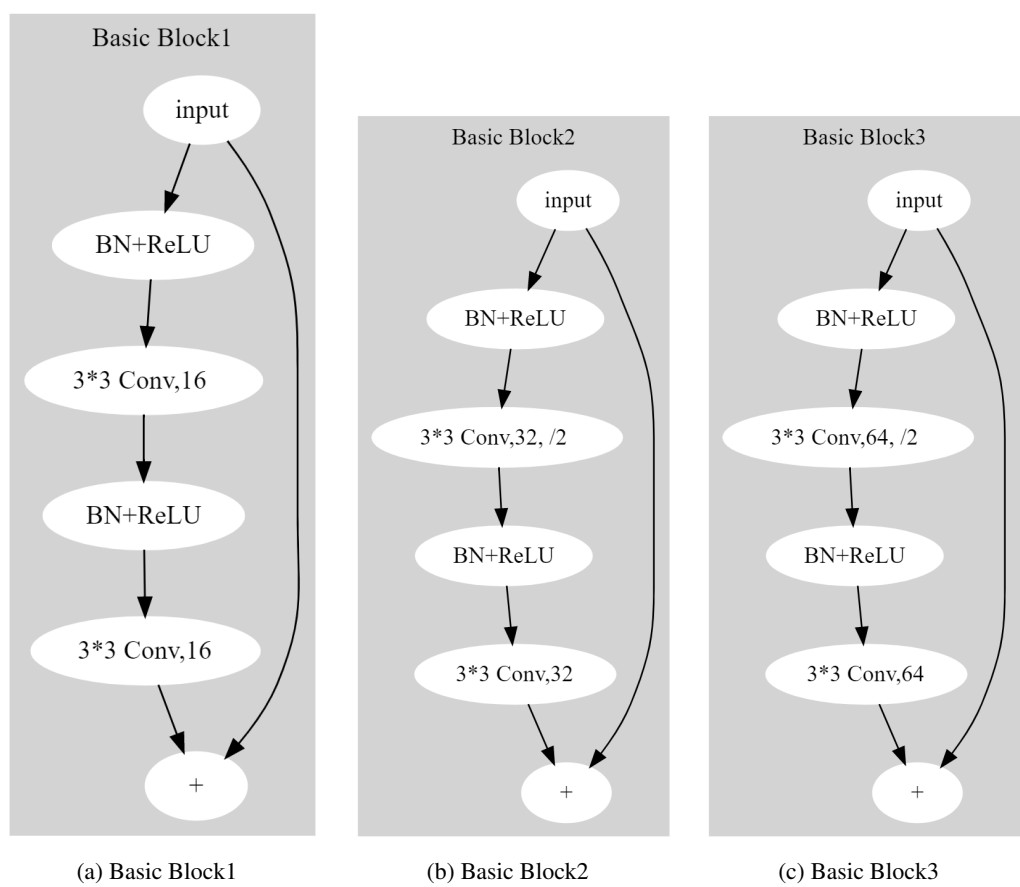

(a) Basic Block1        (b) Basic Block2        (c) Basic Block3

Figure 14: Architectures of the basic building block of ResNets studied in this paper.

