# OpenReview forum: "Deep Learning with Data Privacy via Residual Perturbation"
_ICLR.cc/2021/Conference — Reject_

### Official Review · AnonReviewer3 · 2020-10-25
**Reivew**

**Rating:** 6
**Confidence:** 4

**Review:**

The paper focuses on the topic of differentially private deep learning. Specifically, based on the deep residual learning, they first see it as an ODE. Then, to reduce the reversibility of the ODE, they modify the model as an SDE. By discretizing the SDE, they get a perturbed version of residual learning and use this to design DP-algorithms. The first strategy is directly followed the SDE while the second strategy is with an addition multiplicative noise of the additive noise. Finally, they show that their methods to defend membership inference attack both theoretically and practically.
I tend to accept the paper since I think the paper is well-motivated, also there are privacy guarantees and some theoretical results on defending privacy attack (Theorem 3). However, I still have the following concerns:

1) It is notable that compared with Strategy 1(S1), S2 can only preserve the prediction privacy (since there is and addition multiplicative noise of the additive noise). So in my opinion, comparing the Rademacher complexity between S1 and S2 is unnecessary. So I want to see more comments about this.
2) Moreover, the motivation of S1 is clear which is just followed by the SDE. However, the motivation of S1 is unclear, why the authors add an addition multiplicative noise to the additive noise? Is there any other previous work on it?

---

> ### Author Response · Authors · 2020-11-13
> **Response to Reviewer 3**
>
> Thank you for your valuable feedback and thoughtful reviews. We have revised our manuscript according to your suggestion, and the revised parts are highlighted in blue. Below we address your concerns.
>
> ---
>
> Q1. “It is notable that compared with Strategy 1(S1), S2 can only preserve the prediction privacy (since there is and addition multiplicative noise of the additive noise). So in my opinion, comparing the Rademacher complexity between S1 and S2 is unnecessary. So I want to see more comments about this.”
>
> Reply: First, let us clarify that S2 is independent of S1. In S1, we add additive noise, while we add multiplicative noise in S2 to each residual mapping to protect data privacy. Theoretically, we proved the DP-guarantee for S1 and prediction privacy for S2.
>
> Second, in Theorem 3,  we compared the Rademacher complexity between the continuous analogues of ResNet without noise injection and the ensemble of multiplicative noise-injected ResNets. Rademacher complexity measures overfitting, and lower Rademacher complexity means better generalization bound. We showed that the ensemble of multiplicative noise-injected ResNets can reduce Rademacher complexity, which resonates with our empirical observation that ensemble of noise-injected ResNets can improve classification accuracy (see Appendix D for numerical results). Furthermore, lower Rademacher complexity indicates less overfitting, and overfitting can memorize training data, which breaks data privacy.
>
> ---
>
> Q2. “Moreover, the motivation of S1 is clear which is just followed by the SDE. However, the motivation of S1 is unclear, why the authors add an additional multiplicative noise to the additive noise? Is there any other previous work on it?.”
>
> Reply: We did not add additional multiplicative noise to the additive noise. In our manuscript, we proposed two strategies for protecting data privacy in deep learning. In S1, we injected additive noise into each residual mapping of ResNet. In S2, we inject multiplicative noise into each residual mapping of ResNet. Our empirical results show that multiplicative noise protects better membership privacy than additive noise. There are continuous SDE analogues for both additive noise and multiplicative noise.
>
> Multiplicative noise has been used in other scenarios. For instance, in “X. Liu, T. Xiao, S. Si, Q. Cao, S. Kumar, and C. Hsieh. Neural sde: Stabilizing neural ode networks with stochastic noise.arXiv preprint arXiv:1906.02355.”, the authors used multiplicative noise to improve the robustness of neural networks. We have added this statement to our revised manuscript.
>
>
> =================================
>
> We hope we have cleared your concerns about our work. We have also revised our manuscript according to your comments, and we would appreciate it if we can get your further feedback at your earliest convenience.

---

### Official Review · AnonReviewer2 · 2020-10-28
**Deep Learning with Differential Privacy for ResNets Using SDEs**

**Rating:** 4
**Confidence:** 3

**Review:**

The paper at question tackles the well-known problem of differentially private (DP) deep learning:  already for moderate privacy guarantees, the model performance suffers greatly.

The paper proposes a particular SDE based  method for obtaining privacy for ResNets. DP and stochastic differential equations have been considered in conjunction before e.g. in

Wang, Y.X., Fienberg, S. and Smola, A., 2015, June. Privacy for free: Posterior sampling and stochastic gradient monte carlo. In International Conference on Machine Learning (pp. 2493-2502).

Li, B., Chen, C., Liu, H. and Carin, L., 2019, April. On connecting stochastic gradient MCMC and differential privacy. In The 22nd International Conference on Artificial Intelligence and Statistics (pp. 557-566). PMLR.

(you might consider adding these references).

Although providing an interesting approach by combining DP deep learning and SDEs, I think the paper has some major deficits.
One of them is certain sloppiness with the presentation. I do not understand how the forward and backward Euler discretisations correspond to backward and forward propagation of ResNet layers. The main results are regarding the DP-privacy guarantees of the method (Thm. 1 and Thm. 2).
However these DP-guarantees are not used anywhere in the experiments, and it remains a mystery to me what is actually DP protected.
Another criticism is regarding the experiments: when comparing to DP-SGD, there are no eps,delta-values given, it remains mystery how private the method is and how private is DP-SGD for the choice of hyperparameters listed.

The paper would require a major revision and therefore I cannot recommend it for publication in ICLR.

EDIT: The authors have answered my questions and it clarified a lot. Thus I raise my score by one. However, I am still suspicious about the value of the DP result: for example, it is not discussed why would the residual mapping have an L2-sensitivity G (as stated in the assumptions of the theorem). Also, the reported privacy values in the end of the revised version of the paper (epsilon > 1000) are not meaningful. As far as I see, the experiments give an example where this given membership inference attack works better for DP-SGD protected model than for this residual perturbed version. However the paper does not give any privacy guarantees for the residual perturbation method. Whether the L2-sensitivity can be obtained with e.g. batch normalisation remains unclear to me. I think that the paper would require a careful rewriting.

---

> ### Author Response · Authors · 2020-11-13
> **Response to Reviewer 2**
>
> Thank you for your review. We have added the references you mentioned to our revised manuscript. We believe there are misunderstandings about our paper. Also, we feel that you ignored most of our contributions. Below we address your concerns and restate our major contributions.
>
> ---
>
> Q1. “DP and SDEs have been considered in conjunction before e.g. in
> Wang, Y.X., Fienberg, S. and Smola, A., 2015, June. Privacy for free: Posterior sampling and stochastic gradient monte carlo. ICML (pp. 2493-2502).
> Li, B., Chen, C., Liu, H. and Carin, L., 2019, April. On connecting stochastic gradient MCMC and differential privacy. AISTATS (pp. 557-566). PMLR.
> (you might consider adding these references).”
>
> Reply: Thank you for pointing out these two papers to us, and we have added these to our references. We want to stress that we consider the SDE approach for DP guarantee from a completely different angle from the above two papers. We inject noise into residual mappings, which was motivated by the non-invertibility of SDE. However, the two papers you pointed out considered SDE for DP from the point of the noisy gradient.
>
> ---
>
> Q2. “I do not understand how the forward and backward Euler discretisations correspond to backward and forward propagation of ResNet layers.”
>
> Reply: ResNets are composed of multiple residual blocks that transform the hidden states according to $h_{n+1}=h_n+f(h_n,W_n)$, where $h_n$ is the input, and $f(h_n,W_n)$ is the nonlinear transform parametrized by $W_n$. $h_{n+1}=h_n+f(h_n,W_n)$ can be regarded as Euler discretization for the continuous analogue $\frac{dh(t)}{dt}=f(h(t), W(t))$. In “T. Chen, Y. Rubanova, J. Bettencourt, and D. Duvenaud. Neural ordinary differential equations. NeurIPS, 2018.”, the authors leveraged this continuous viewpoint of ResNets to propose Neural ODEs.
>
> We use the backward Euler discretization as a way to recover the initial data from the terminal value obtained from the forward Euler discretization, which is an illustration of reverse-engineering input from the output of machine learning models. We did not model backward propagation of ResNets as backward Euler discretization.
>
> In our paper, we use the continuous analogy of ResNets to understand data-privacy issues in deep learning and propose two privacy-protection strategies.
>
> ---
>
> Q3. “The main results are regarding the DP-privacy guarantees of the method (Thm. 1 and Thm. 2). However these DP-guarantees are not used anywhere in the experiments, and it remains a mystery to me what is actually DP protected. Another criticism is regarding the experiments: when comparing to DP-SGD, there are no eps, delta-values given, it remains mystery how private the method is and how private is DPSGD for the choice of hyperparameters listed.”}
>
> Reply: Please allow us to clarify your concerns on our contribution, DP budget, and DPSGD hyperparameters issues.
>
> First, beyond the privacy guarantees in Thm. 1, Thm. 2, and generalization bound in Thm. 3. We proposed two very simple and computationally efficient algorithms for protecting data privacy in deep learning. Moreover, we empirically verified the classification accuracy and membership privacy.
>
> Second, our proposed algorithm guarantees DP, but the current DP budget is not as tight as DPSGD.  We have added privacy budget comparisons between residual perturbation and DPSGD in section 4.6, and there are several difficulties we need to overcome to get tight DP bounds for residual perturbation. Compared to DPSGD, it is significantly harder. In particular, 1) the loss function of the nose injected ResNets is highly nonlinear and very complex with respect to the weights, also the noise term appears in the loss function due to the noise injected in each residual mapping. These together make the tight estimate very difficult. 2) In our proof, we leveraged the framework of R\’enyi-DP (RDP) to find a feasible range of noise variance parameter $\gamma$, and then convert to DP to get the value of $\gamma$ for a given DP budget. This procedure will significantly reduce the accuracy of the estimated $\gamma$. We leave the tight DP guarantee as future work. In particular, how to reduce the accuracy of estimating $\gamma$ due to the conversion between RDP and DP. We believe it is reasonable, considering there are efforts from many researchers to get the current DP-budget for DPSGD.
>
> Third, empirically, we have remarkable advantages in terms of membership privacy protection, utility enhancement, and computational efficiency over DPSGD. Regarding hyper-parameters of DPSGD, we have tried our best to tune all related hyper-parameters in the Tensorflow DPSGD benchmark  (https://github.com/tensorflow/privacy) to get the reported tradeoff between membership privacy and classification accuracy.
>
> ===
>
> We hope we have cleared your concerns about our work. We have also revised our manuscript according to your comments, and we would appreciate it if we can get your further feedback at your earliest convenience.

---

### Official Review · AnonReviewer4 · 2020-11-06
**Interesting ideas but results seem preliminary.**

**Rating:** 6
**Confidence:** 4

**Review:**

### Summary

The paper presents a method for training ResNets with differential privacy. Rather than the usual methods based on noisy gradient descent, the authors propose adding noise at each layer of the network during both training and testing. The authors prove differential privacy guarantees for two strategies of this type (one with additive and one with multiplicative noise). They also show some evidence that the noise can help generalization, by showing that the Rademacher complexity of a continuous linearized version of the model is lower when noise is added.

### Evaluation: Theory

The theoretical results are a start, but have some limitations which seem significant to me:

* Theorem 1 needs the output of any residual mapping to be bounded in expectation. It is not clear whether this actually holds.

* Theorem 2 only gives privacy guarantees for a single prediction. This is interesting but not as a strong as outputting the model.

* Theorem 3 is about a continuous linearized analogue of the models. It also does not have quantitative bounds on how much the Rademacher complexity can improve.

### Evaluation: Experiments

The experiments are promising: the authors run a membership inference attack against their models and against a model trained with DPSGD, to test privacy. They also test accuracy. I am, however, bothered that there is no attempt to give tight differential privacy bounds and to compare the two algorithms with choices of parameters that give the same provable differential privacy bounds. The membership inference attack is supposed to be a proxy for that but I am not convinced that just running one attack and getting a slightly lower AUC is good evidence that the proposed model preserves privacy as strongly as DPSGD. In general, I do not want to see work on privacy in ML adopt the strategy of running a single membership inference attack to verify privacy. This is not convincing: what if a slightly different attack does a lot better? The point of rigorous privacy guarantees is that they hold against all attacks.

---

> ### Author Response · Authors · 2020-11-13
> **Response to Reviewer 4**
>
> Thank you for your valuable feedback and thoughtful reviews. We have revised our manuscript according to your suggestion, and the revised parts are highlighted in blue. Below we address your concerns.
>
> ------
>
> Q1. “Theorem 1 needs the output of any residual mapping to be bounded in expectation. It is not clear whether this actually holds.”
>
> Reply: There is a batch normalization at the end of each residual mapping, which makes our assumption on the boundedness of the expectation of each output of residual mapping reasonable.
>
> -----
>
> Q2. “Theorem 2 only gives privacy guarantees for a single prediction. This is interesting but not as strong as outputting the model.”
>
> Reply: We agree with your point that the privacy guarantee provided in Theorem 2 is not as strong as outputting the model. Theorem 2 guarantees the privacy of the training data given only black-box access to the model, i.e., the model will output the prediction for any input without granting adversaries access to the model itself. In particular, we cannot infer whether the model is trained on $S$ or $S'$ no matter how we query the model in a black-box fashion. For the second strategy, the theoretical DP-guarantee for the whole model is under our study, which is technically much harder than the proof for Theorem 1. We have made this point clear in the revised manuscript.
>
> ------
>
> Q3. “Theorem 3 is about a continuous linearized analogue of the models. It also does not have quantitative bounds on how much the Rademacher complexity can improve.”
>
> Reply: In the proof of Theorem 3, which is available in Appendix C, we have provided detailed bounds of the Rademacher complexity of different models. In particular, the ODE analogue of ResNet without noise injection has a Rademacher complexity
>
> $$(c/N)\exp(cTp)\mathbb{E}_\sigma\|\sum_{i=1}^N\sigma_i x_i^p\|_2,$$
>
> while the SDE analogue of the ensemble of noise injected ResNets has a Rademacher complexity
>
> $$(c/N) \exp(cTp-p(1-p) \gamma^2 T/2)\mathbb{E_\sigma\|\sum_{i=1}^N\sigma_i x_i^p\|_2,$$,
>
> where $p\in (0, 1)$ and the meanings of other notations can be found in our manuscript. Therefore, it is easy to find the improvement in terms of Rademacher complexity.
>
> ------
>
> Q4. “I am, however, bothered that there is no attempt to give tight differential privacy bounds and to compare the two algorithms with choices of parameters that give the same provable differential privacy bounds.”
>
> Reply: As discussed in section 4.6, there are several difficulties we need to overcome to get tight DP bounds for our proposed strategies. Compared to DPSGD, it is significantly harder. In particular, 1) the loss function of the nose injected ResNets is highly nonlinear and very complex with respect to the weights, also the noise term appears in the loss function due to the noise injected in each residual mapping. These together make the tight estimate very difficult. 2) In our proof, we first leveraged the framework of R\’enyi-DP (RDP) to find a feasible range of noise variance parameter $\gamma$, and then convert to DP to get the value of $\gamma$ for a given DP budget. This procedure will significantly reduce the accuracy of the estimated $\gamma$. We have added privacy budget comparisons between residual perturbation and DPSGD in section 4.6.
>
> Empirically, we have shown remarkable advantages in terms of privacy protection, utility enhancement, and computational efficiency over DPSGD. Theoretically, we can also provide the DP guarantee for the proposed Strategy I. We leave the tight DP guarantee as future work. In particular, how to reduce the accuracy of estimating $\gamma$ due to the conversion between RDP and DP. We believe it is reasonable, considering there are efforts from many researchers to get the current DP-budget for DP-SGD.
>
> ------
>
> Q5. “The membership inference attack is supposed to be a proxy for that but I am not convinced that just running one attack and getting a slightly lower AUC is good evidence that the proposed model preserves privacy as strongly as DPSGD. In general, I do not want to see work on privacy in ML adopt the strategy of running a single membership inference attack to verify privacy. This is not convincing: what if a slightly different attack does a lot better? The point of rigorous privacy guarantees is that they hold against all attacks.”
>
> Reply: The membership inference attack is a benchmark attack to evaluate privacy-protection. Differential privacy provides a better theoretical privacy guarantee against possible attacks. In our manuscript, we have shown the DP guarantee for the proposed Strategy I.
>
>
> ========================
>
> We hope we have cleared your concerns about our work. We have also revised our manuscript according to your comments, and we would appreciate it if we can get your further feedback at your earliest convenience.

---

### Official Review · AnonReviewer6 · 2020-11-10
**Interesting topic but there remains some concerns on the utility enhancement the DP guarantee**

**Rating:** 5
**Confidence:** 3

**Review:**

Summary:
This paper studies an important problem and proposes the novel residual perturbation to protect privacy while maintaining the ResNet models’ utility.  Two SDE models are provided to inject noises with abundant theoretical proof are provided. Experimental results demonstrate the performance of privacy protection and classification accuracy on benchmark datasets. My major concern is about the utility enhancement and the DP guarantee (see cons below). Hope the authors can address my concern in the rebuttal period.

Pros:
1. The paper studies a fundamental problem on how to protect privacy while retaining model utility. The problem itself will have great impacts on real-world scenarios.
2. The proposed two strategies are novel for injecting noise to each residual mapping of ResNet theoretically principled by the stochastic differential equation theory. Great amounts of theoretical proof are provided and seem technically sound.
3. Experiments on the real-world dataset provide some interesting insights about the advantages of residual perturbation.

Cons:
1. Experiments demonstrate that utility is enhanced after noise injection. In general, there is a tradeoff between privacy and utility. This paper can increase both, and the authors owe this enhancement to the ensemble of noise injected ResNets. It seems not very clear to me that how to conduct the ensemble. What’s more, if it is just because of the ensemble, what is the contribution of this paper?
2. There should be proof that each iteration of the whole model by strategy I and II can satisfy DP, while authors only prove the parameters can satisfy DP.

Questions during the rebuttal period: Please address and clarify the cons above.

---

> ### Author Response · Authors · 2020-11-13
> **Response to Reviewer 6**
>
> Thank you for your valuable feedback and thoughtful reviews. We have revised our manuscript according to your suggestion, and the revised parts are highlighted in blue. Below we address your concerns.
>
> -------
>
> Q1. “Experiments demonstrate that utility is enhanced after noise injection. In general, there is a tradeoff between privacy and utility. This paper can increase both, and the authors owe this enhancement to the ensemble of noise injected ResNets. It seems not very clear to me that how to conduct the ensemble. What’s more, if it is just because of the ensemble, what is the contribution of this paper?”
>
> Reply: We conduct the model ensemble as follows: Suppose we have n ResNets with noise injected in each residual mapping and without weights sharing. Then, we average the output of these n models, which gives the prediction. During training, we back-propagate loss and jointly train these n models. This ensemble is the same as that used in “B. Wang, Z. Shi, and S. Osher. ResNets ensemble via the Feynman-Kac Formalism to Improve Natural and Robust Accuracies. NeurIPS, 2019.”
>
> Regarding your concerns about the privacy-utility tradeoff of our work, on the one hand, the injected noise in each residual mapping helps to protect data privacy. On the other hand, the model ensemble without weights sharing increases model capacity, which can help to improve the utility of the model. Meanwhile, intuitively the injected noise can regularize the model and reduce overfitting.
>
> Let us clarify our contributions below, which are threefold. 1) We proposed two very simple residual perturbations to protect data privacy, which was inspired by the continuous analog of ResNet. 2) We analyzed the theoretical privacy and generalization bound of the proposed residual perturbation. 3) Numerically, we verified the efficiency of the proposed residual perturbations in terms of classification accuracy and membership privacy protection. Compared to the benchmark DP-SGD, the residual perturbation is computationally more efficient, which does not need to compute the per-sample gradient. Moreover, residual perturbation degrades less utility and protects better membership privacy than the benchmark DP-SGD.
>
>
> -------
>
> Q2. “There should be proof that each iteration of the whole model by strategy I and II can satisfy DP, while authors only prove the parameters can satisfy DP.”
>
> Reply: The DP guarantee for parameters of the model is commonly used in the community. For instance, it is used in “K. Chaudhuri, C. Monteleoni, and A. Sarwate. Differentially Private Empirical Risk Minimization. JMLR, 2011.”
>
> Theorem 1 in our revised manuscript guarantees DP for each iteration of the whole model through residual perturbation by using Strategy I. The privacy guarantee strategy II can be found in Theorem 2, which intuitively protects the membership privacy for the training data.
>
>
> =================================
>
> We hope we have cleared your concerns about our work. We have also revised our manuscript according to your comments, and we would appreciate it if we can get your further feedback at your earliest convenience.

---

### Decision · Program_Chairs · 2021-01-07
**Final Decision**

**Decision:**

Reject

**Comment:**

This paper proposes techniques for differentially private training of ResNets inspired by SDEs. The idea has some promise but the paper does not give convincing evidence, either theoretical or empirical that it outperforms existing tehniques. Unfortunately, comparisons with existing techniques are presented in a misleading way that does not clearly provide the privacy parameters.
Another issue with the work is that the authors appear to be unaware of (and hence do not compare with) existing work on privacy of a single prediction (referred to as strategy II in this work).
Approaches for this problem are described in these theoretical works (and several follow ups)

* Dwork, Feldman. Privacy-preserving Prediction. COLT 2018
* Bassily,Thakkar,Tahkurta. Model-Agnostic Private Learning via Stability. NIPS 2018

Practical results are mentioned in PATE papers of Papernot et al. and more recent work
* van der Maaten,  Hannun. The Trade-Offs of Private Prediction https://arxiv.org/abs/2007.05089